# The More the Merrier? Zoo Visitors' Affective Responses and Perceptions of Welfare across an Increase in Giraffe Density

Wilson C. Sherman [1,*], Darren E. Minier [2], Caterina N. Meyers [2] and Michelle L. Myers [2]

1   Environmental Science, Policy and Management, University of California, Berkeley, CA 94720, USA
2   Conservation Society of California—Oakland Zoo, Oakland, CA 94605, USA;
    dminier@oaklandzoo.org (D.E.M.); cmeyers@oaklandzoo.org (C.N.M.); mmyers@oaklandzoo.org (M.L.M.)
*   Correspondence: wilsonsherman@berkeley.edu

**Abstract:** Zoos strive to create experiences that inspire positive feelings toward animals which lead to conservation behaviors in their visitors. However, concerns regarding the welfare of animals living in zoos present a challenge in creating positive zoo experiences and promoting the conservation agenda and moral authority of these cultural institutions. This research explores connections between zoo visitors' positive affective responses and their assessments of animal welfare before and after two giraffes were introduced to a group of four giraffes in a multi-species savannah exhibit. A self administered questionnaire was completed by 499 visitors to the Conservation Society of California's Oakland Zoo. The questionnaire measured visitors' predispositions, affective responses, and assessments of animal welfare. Results suggest that visitors' assessments of animal welfare, positive affective experience, and predisposition are positively correlated. Further, visitor assessments of animal welfare are generally more positive after the addition of new giraffes. Although visitors tended to report that the giraffes were very healthy and well cared for, they responded less positively when asked about how happy the giraffes were and how adequately sized their exhibit was. The findings suggest that understanding and improving zoo visitors' assessments of animal welfare is important in improving positive experiences and conservation education outcomes during a visit to the zoo.

**Keywords:** perceptions of animal welfare; affective experience; visitor studies; zoos and aquariums

## 1. Introduction

As cultural institutions dedicated to the conservation of biodiversity, zoos and aquariums aim to provide experiences that inspire connection to nature and conservation action in their guests [1] With more than 700 million annual visitors globally [2], zoos and aquariums (hereafter zoos) are important settings where discourse around animals, conservation, and welfare contribute to collective meaning making in the "culture of nature" [3]. As zoos seek to fulfil missions of conservation and education, understanding factors that contribute to the learning and satisfaction of zoo guests is increasingly relevant.

In studies of affect in zoo visitors, research has found that the positive emotions felt while observing zoo animals are highly relevant in processes of learning and meaning making. The affective domain encompasses states of emotion which can lead to moods and sentiments over time; zoo visitors report experiencing a breadth of positive emotions, such as a sense of beauty, respect, and wonder while observing animals [4]. The positive feelings resulting from observations of animal behavior are the basis for more complex reactions to the zoo experience, including increased implicit connectedness to nature, empathy for animals, interest in learning, and motivation to engage in pro-environmental behaviors [4–11].

Moral issues regarding the housing of live animals are an intense part of the critique of zoos as cultural institutions and could threaten their conservation potential and commercial viability [1,12]. The perceived welfare of zoo animals influences visitor satisfaction at

individual zoos as well as public trust in these institutions as a whole [12–17]. Zoo visitors make judgements about animal welfare based on observations of animal behavior and the exhibit in which the animal lives, often in comparison with the behaviors and environments expected of animals in nature [18]. Packer et al. [19] studied visitor perceptions of the welfare of gorillas living at the Brookfield Zoo and found that zoo visitors assessed animal happiness based on observed behavior, assessed animal health based on the animal's physical condition, and assessed the quality of the care they received based on judgments of the animals' environment. Miller et al. [20] found that visitor's perceptions of elephant welfare were correlated with their perceptions of the exhibit and that visitors value exhibits that are perceived as large enough, well maintained, and natural. Godinez et al. [21] found that animal behavior influenced guest perceptions of a jaguar exhibit, as visitors reported finding the exhibit less adequate for the animal's needs when a jaguar was engaged in a stereotypic pacing behavior. Perceptions of the relationship between animal care staff and animals may also contribute to welfare perceptions. In a study of guest reactions to ambassador animal presentations, Minarchek et al. [22] found that when watching a presentation where animal handlers gave armadillos more choice and control (animals could choose to come out of their enclosure and were not held or touched), guests had more positive perceptions of welfare when compared to presentations in which handlers used traditional handling methods (animals were removed from their enclosure and handled). Despite a tendency for zoo professionals to disregard non-expert assessments of animal welfare, Veasey [12] found a strong correlation between holistic assessments of animal welfare by animal care staff and assessments of animal happiness by zoo visitors.

Guest perceptions of animal welfare have been found to be positively correlated with learning outcomes, conservation intent, positive emotional experiences, emotional connection with animals, and empathic reactions [19,20,22]. Given the role that perceptions of welfare play in the zoo experience, studies of visitor satisfaction and learning that ignore perceptions of animal welfare may be incomplete or misleading [23].

This study was designed to understand the connection between animal welfare perceptions and positive affective experience as well as to study the role of animal density in perceptions of welfare. This study focused on giraffes, as giraffes display relatively consistent levels of activity and visibility, to control for the influence of activity and visibility on visitor reaction [10,24–26].

The study aims to answer the following research questions:

(1) Does visitor perception of giraffe welfare relate to the visitor's affective response to observing the animals?
(2) Do visitor's predispositions toward wildlife relate to their perceptions of welfare?
(3) How will the introduction of new giraffes and an increase in giraffe density in the exhibit influence guest welfare perceptions?

## 2. Methods

### 2.1. Overview

In spring (March–April) and summer (June–July) of 2021, self-administered questionnaires were distributed to adult visitors to the Conservation Society of California's Oakland Zoo African Veldt exhibit during periods before ($n$ = 269) (hereafter "phase one") and after ($n$ = 230) (hereafter "phase two") the introduction of two reticulated giraffes to an existing group of four giraffes. The instrument measured visitors' predispositions to nature, positive affective responses, and assessments of animal welfare.

### 2.2. Study Site

The Oakland Zoo is accredited by the Association of Zoos and Aquariums and situated on 100 acres in Oakland, CA, USA. The African Veldt exhibit is 25,000 square feet and includes a waterfall and moat. The exhibit features signs displaying information about the natural history and conservation of the species displayed. During the first period of the study, the exhibit housed four reticulated giraffes (*Giraffa reticulata*), three common

eland (*Taurotragus oryx*), and two Egyptian geese (*Alopochen aegyptiaca*). The giraffe herd consisted of two female giraffes, ages 9 and 2, and two male giraffes, ages 22 and 14. The second period of the study began after the introductions of two additional male giraffes, aged 4 and 1. The second phase of the study began 1 month after the giraffe introductions began, when animal care staff reported that all six giraffes would consistently be on exhibit.

### 2.3. Procedures

A convenience sample of visitors was recruited by research assistants. Visitors to the exhibit were selected using a modified version of the next-across-the-line approach in which the next group of visitors to stop in a section of the giraffe exhibit were invited to participate in the survey. Groups in which young children outnumbered adults were excluded because managing children can prevent adults from thoughtful participation [7]. Visitors filled out the survey on an iPad using Survey Monkey Anywhere [27]. Visitors completed the survey on their own unless the participant requested that the researcher read questions aloud. Visitors were asked not to collaborate with others in their party. Respondents generally took 3–5 min to complete the survey. Data for phase one were collected between March and April of 2021, and data for phase two were collected between June and July of the same year.

### 2.4. Instrument

The instrument included questions measuring visitors' predispositions, affective responses, and animal welfare perceptions. Predispositions were measured using a slightly adapted version of Brookfield Zoo's Internal Visitor Predisposition Scale [10], which was designed to assess visitors' interest in mission related subjects, including animals, the environment, and conservation behavior. This scale has been tested in multiple zoos, is factor free, and consistently displays a high level of internal consistency [10]. To measure guests' affective experiences, the instrument asked guests to rate the intensity of 7 positive feelings that might be experienced while viewing the giraffe on a 7-point scale [4,10,28,29]. The animal welfare perception scale used items from Packer et al. [19] where visitors are asked to rate how happy, healthy, and well cared for the giraffes are; items asking how appropriate the size of the exhibit is, and how adequate the giraffes' social group was were added. These questions were rated on a 10-point scale. Finally, age, gender, and zoo visitation demographics were collected. Incomplete responses, responses accidentally collected by those under the age of 18, and responses that were collected during periods when individual giraffes were kept off exhibit were removed from the sample.

### 2.5. Statistical Analysis

Data were analyzed using R [30], using packages "tidyverse" [31] "psych" [32], and "likert" [33]. The figure was designed using Adobe Illustrator [34]. Likert-type data violates assumptions of parametric tests of significance because it is discrete and abnormally distributed. Although with large sample sizes, parametric and non-parametric tests show similar strength, we used non-parametric tests of significance because the data were ordinal and not normally distributed. Mann–Whitney U Tests were used for variables with two groups and Kruskal–Wallis tests were used for variables with more than two groups. A Bonferroni adjustment was used to correct for an increased Type I error rate across multiple comparisons. Adjusted *p*-values < 0.05 were considered significant.

## 3. Results

### 3.1. Sample Description

The sample was 55% female, 41% male, and 4% non-binary. Furthermore, 47% of the sample was 18–35 years old, 40% were 36–55 years old, 11% were 56–75 years old, and <1% were older than 76 years old. Sixty-three percent of respondents had visited the Oakland Zoo before and 31% were members of the Oakland Zoo. Forty-three percent of the sample

reported visiting zoos more than once a year, 26% reported visiting zoos once a year, and 31% reported visiting zoos less than once a year.

### 3.2. Scale Reliability

Principal component analysis using polychoric correlation was used on each scale to determine that the three scales were factor free, measuring a single underlying construct. Cronbach's alpha scores were calculated to further test the reliability of the scales for the visitor predisposition scale $\alpha = 0.83$, the affective experience scale $\alpha = 0.87$, and the perceptions of animal welfare scale $\alpha = 0.80$. Cronbach's alpha scores would not improve if any items from the scales were removed, according to item reliability analyses. Composite scores for each construct were calculated for each respondent.

### 3.3. How Are These Scales Related?

In order to understand relationships between scales, Kendall's tau correlations were computed including data from both phases. A moderate and statistically significant positive correlation (using Kendall's Tau correlation) was found between visitor predisposition and affect scores ($r\tau = 0.35$, $p < 0.001$) as well as between affect scores and welfare scores ($r\tau = 0.25$, $p < 0.001$). A weak but statistically significant positive correlation was found between predisposition scores and welfare scores ($r\tau = 0.17$, $p < 0.001$). When affective experience is held constant using partial correlation, correlation between visitor predispositions and perceptions of welfare drops to $r\tau = 0.09$. When visitor predispositions are held constant, the correlation between the perception of animal welfare score and affective score drops to $r\tau = 0.21$. When welfare scores are held constant, the correlation between predisposition and affective score drops to $r\tau = 0.32$. This suggests that although a visitor's perception of animal welfare is related to their affective experience viewing animals, their predisposition to nature is a stronger predictor of affective experience. Further, responses to the visitor predisposition to nature scale are not strongly correlated with the perception of welfare.

### 3.4. Scale Performance across the Giraffe Introduction

Difference in item and scale means before and after the introduction of the new giraffes are presented in Table 1. A significant difference was found between the phases in one of eight items in the predisposition scale, with visitors in phase two agreeing more with the statement "you are ordinarily interested in animals". No significant differences were found between means in any item in the affective scale. Of the items regarding perceptions of welfare, a significant difference between phases was found in the item asking guests to describe the adequacy of the giraffes' social group size (Figure 1).

**Table 1.** Mean responses to survey questions before and after the introduction of new giraffes. *p* values < 0.05 *; *p* values < 0.005 **.

| How Much Do You Agree with the Following on a 7 Point Scale from 1 (*Not at All*) to 7 (*Very Much So*) | Visitor Predisposition Scale | | | | | |
| --- | --- | --- | --- | --- | --- | --- |
| | Phase One | | Phase Two | | | |
| | Mean | SD | Mean | SD | Test Statistic | *p*-Value |
| You are ordinarily interested in animals | 6.05 | 1.15 | 6.34 | 1.07 | 27,291 | 0.0456 * |
| You often feel a sense of connection with nature | 5.86 | 1.35 | 5.86 | 1.27 | 32,878 | 1 |

**Table 1.** *Cont.*

| How Much Do You Agree with the Following on a 7 Point Scale from 1 (*Not at All*) to 7 (*Very Much So*) | Visitor Predisposition Scale | | | | | |
| | Phase One | | Phase Two | | | |
| | Mean | SD | Mean | SD | Test Statistic | *p*-Value |
| You have a good understanding of wildlife conservation issues | 4.7 | 1.46 | 4.79 | 1.49 | 31,184 | 1 |
| You usually try to help protect and preserve local wildlife habitats | 5.23 | 1.55 | 5.5 | 1.57 | 28,808 | 1 |
| You pay attention to news about environmental issues | 5.01 | 1.5 | 5.34 | 1.45 | 28,213 | 0.527 * |
| You tend to support conservation organizations (volunteer your time, make a donation, sign a petition, etc.) | 4.03 | 1.84 | 4.4 | 1.77 | 28,556 | 0.994 |
| You typically engage in conservation efforts during your daily activities (recycling, reducing your energy usage, buying earth friendly products, etc.) | 5.47 | 1.44 | 5.66 | 1.46 | 29,635 | 1 |
| You spend as much time as you can in natural settings such as parks and open spaces, beaches, and lakes | 5.6 | 1.37 | 5.52 | 1.34 | 33,851 | 1 |
| Composite Score | 5.24 | 0.98 | 5.43 | 0.97 | 27,131 | 0.674 |
| | Affective Response | | | | | |
| Rate the Intensity of the Following Feelings Experienced While Observing the Giraffes from 1 (*Not at All*) to 7 (*Very Much So*) | Mean | SD | Mean | SD | Test Statistic | *p*-Value |
| Curiosity | 5.67 | 1.28 | 5.77 | 1.3 | 30,788 | 1 |
| Respect/Admiration | 6.36 | 0.94 | 6.33 | 1.09 | 31,744 | 1 |
| Wonder/Awe | 6.02 | 1.25 | 6.1 | 1.18 | 31,396 | 1 |
| Amusement | 5.44 | 1.56 | 5.44 | 1.51 | 32,617 | 1 |
| Sense of Connection | 4.91 | 1.57 | 4.96 | 1.56 | 31,629 | 1 |
| Love | 5.5 | 1.44 | 5.57 | 1.52 | 31,015 | 1 |

**Table 1.** *Cont.*

| How Much Do You Agree with the Following on a 7 Point Scale from 1 (*Not at All*) to 7 (*Very Much So*) | Visitor Predisposition Scale | | | | | |
|---|---|---|---|---|---|---|
| | Phase One | | Phase Two | | | |
| | Mean | SD | Mean | SD | Test Statistic | *p*-Value |
| Attraction | 5.33 | 1.58 | 5.3 | 1.6 | 32,700 | 1 |
| Composite Score | 5.6 | 1.03 | 5.64 | 1.06 | 29,971 | 1 |
| **Animal Welfare Assessment** | | | | | | |
| Question | Mean | SD | Mean | SD | Test Statistic | *p*-Value |
| Overall how happy do the giraffes appear to you on a scale of 1 to 10 where 1 is very unhappy and 10 is very happy? | 7.16 | 2.07 | 7.57 | 2 | 28,783 | 1 |
| Overall how healthy do the giraffes appear to you on a scale of 1 to 10 where 1 is very unhealthy and 10 is very healthy? | 8.57 | 1.57 | 8.85 | 1.59 | 28,446 | 0.395 |
| Overall how well cared for do the giraffes appear to you on a scale of 1 to 10 where 1 is very poorly cared for and 10 is very well cared for? | 8.56 | 1.58 | 8.86 | 1.48 | 28,931 | 0.843 |
| Overall, how adequate would you say the size of the exhibit is for these animals on a scale of 1 to 10 where 1 is cramped and 10 is spacious? | 6.23 | 2.64 | 6.38 | 2.6 | 30,805 | 1 |
| Overall, how would you describe the size of the giraffes' social group on a scale of 1 to 10 where 1 is inadequate and 10 is adequate? | 7.28 | 2.12 | 7.97 | 2 | 26,355 | 0.006 ** |
| Composite Score | 7.56 | 1.5 | 7.93 | 1.48 | 25,816 | 0.054 |

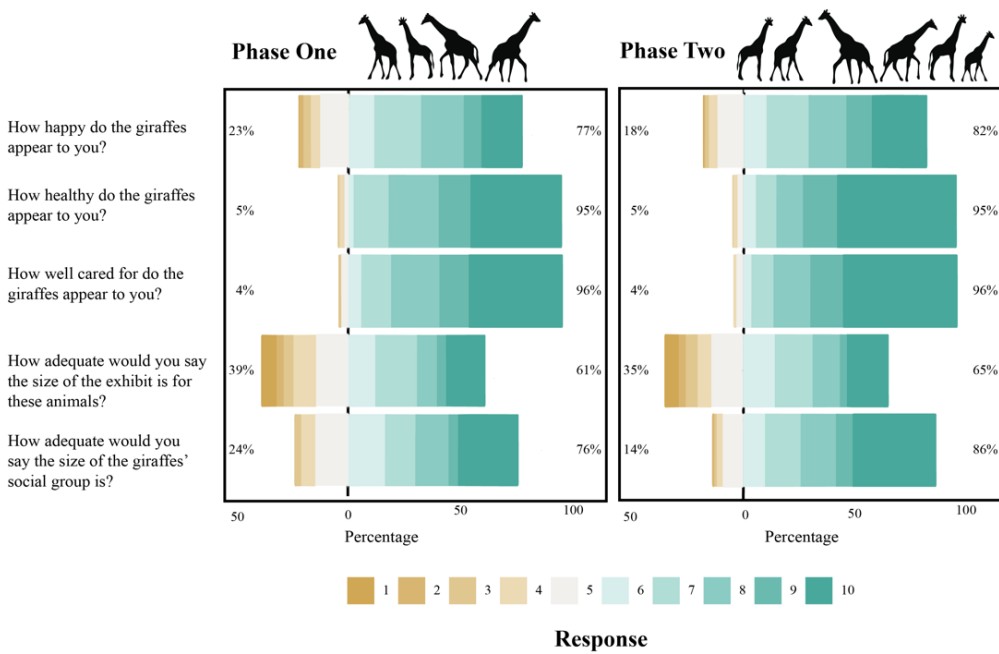

**Figure 1.** Animal welfare perception scale response by phase.

### 3.5. Perceptions of Welfare across Demographics

Differences in mean composite scores for visitor predisposition, affective reaction, and perception of animal welfare across demographic groups are presented in Table 2. A significant difference was found between affect scores ($p < 0.005$) across gender, with women averaging higher than men (Table 2). Significant differences were found in predisposition scores ($p < 0.00001$) across age, with people in older age groups having higher predisposition scores (Table 2). Although the adjusted $p$-value for difference between welfare scores across age was just above the significance level, people in older age groups had higher mean welfare scores than the people in younger age groups (Table 2).

**Table 2.** Mean predisposition, positive affect, and animal welfare scores across demographics. $p$-values < 0.005 **, and $p$-values < 0.0005 ***.

|  | Mean Predisposition Score (Out of 7) | Mean Affect Score (Out of 7) | Mean Welfare Score (Out of 10) |
|---|---|---|---|
| **Results by Gender** | | | |
| **Male** | 5.24 | 5.36 | 7.58 |
| **Female** | 5.41 | 5.79 | 7.88 |
| **Other** | 5.02 | 6.23 | 7.64 |
| **Chi-Squared** | 5.47 | 23.64 | 10.889 |
| **Degrees of Freedom** | 4 | 4 | 4 |
| **$p$-Value** | 1 | 0.003 ** | 1 |
| **Results by Age Group** | | | |
| **18–35** | 5.21 | 5.6 | 7.57 |
| **36–55** | 5.29 | 5.61 | 7.72 |
| **56–75** | 5.93 | 5.67 | 8.37 |
| **76+** | 5.97 | 6.25 | 8.55 |
| **Chi-Squared** | 29.985 | 29.985 | 15.561 |
| **Degrees of Freedom** | 3 | 3 | 3 |
| **$p$-Value** | <0.001 *** | 1 | 0.053 |

## 4. Discussion

Our study was designed to investigate the relationship between zoo visitor perceptions of animal welfare and their affective experience while also testing the influence of a change in giraffe density on welfare perceptions. After the introduction of the two new giraffes, mean scores for every item on the scale assessing welfare increased. However, the only welfare item which showed a statistically significant difference was the item asking about the adequacy of the giraffes' social group size. This result indicates that the change in giraffe herd size was noticeable and perceived as contributing positively to the giraffes' welfare.

We found that guests perceived that the giraffes were very healthy and well cared for but just above the midpoint of the scale in terms of happiness and adequate exhibit size. Of interest is the gap in perceptions between how "well cared for" the giraffes seemed and how "happy" they seemed. Packer et al. [19] found a similar divide between perceived animal happiness and the perceived quality of their care. This gap may be explained by a fundamental belief that an animal can only be so happy in captivity, but the perceived inadequacy of the exhibit size could also be a contributing factor (in their study of public trust in zoos and aquariums, Rank et al. [15] found that the largest gap between perceived current performance of zoos and expectation for establishing trust was an item asking if zoo exhibits have adequate space to meet the needs of their animals). Despite relatively low scores for the adequacy of exhibit size, the increased number of giraffes in the space was positively received. This may indicate that exhibit size is judged based on animal size or expectations of the animal's natural habitat rather than a square foot per animal calculation. The appearance of the increased opportunity for socialization or the presence of a young giraffe and the subsequent appearance of family structure could also explain this increase in perceived animal welfare.

Another aim of this study was to better understand the connection between zoo visitors' perceptions of animal welfare and their affective experiences while observing giraffes at the zoo. Our results suggest that zoo visitor's positive affective experiences are related to their perceptions of animal welfare, with visitors who perceived the giraffes to be in a better state of welfare having more positive emotional reactions. This result supports research which emphasizes the importance of guests' perceptions of welfare in mission related learning and meaning making during a visit to the zoo. No significant differences were seen between affect scores across the two phases of the study, indicating that although guests did notice the increase in the giraffe herd size and perceive it as positively influencing giraffe welfare, it did not dramatically alter their affective experiences. Across the two phases of the study, the only significant difference between visitor predispositions out of seven items was the item asking guests how much they agreed with the statement "I am ordinarily interested in animals". Perceptions between the groups were otherwise relatively consistent.

Our study found a stronger relationship between the affective experience of zoo guests and their self-reported predispositions to nature when compared to the relationship found between affective experience and perception of welfare. Other studies have also found correlations between a visitor's affective or empathic reaction and their predisposition to nature [10,22]. These results bring up concerns voiced in previous research about which parts of their audience zoos serve most effectively: is a visit to the zoo stimulating positive emotional responses and connection to nature in all guests or only in those who already identify with and prioritize nature (also known as preaching to the choir) [9,35]?

Something else worth noting in our results is the weak relationship between visitor predisposition toward nature and perceptions of welfare, which was also found in Minarchek et al. [22], as well as the result that perceptions of animal welfare did not vary significantly by zoo membership or frequency of zoo visitation. These results were contrary to our initial hypotheses that visitors who scored higher on the predisposition scale would be more critical of animal welfare due to a greater connection to and respect for nature (an expected negative relationship) and that members or frequent zoo visitors would have more positive views of welfare because they have demonstrated support for zoos as institutions.

Although our results indicated that women and people who were older than 36 gave higher animal welfare scores, future research should seek to identify which preexisting beliefs or value orientations amongst zoo guests are correlated with their perceptions of welfare, in order to identify which groups of zoo visitors are more likely to have their experience and learning outcomes diminished by concerns about the welfare of the animals in the zoo. Dietz et al. [36] explore similar themes, pointing out that a "concern for animals" value orientation constitutes a distinct motivation from other values traditionally used in environmental decision making research, in which people with a "concern for animals" value orientation had a higher identification with animal rights issues than people with "biospheric" value orientations. The opposite was found to be true regarding identification with the environmental movement.

*Limitations*

This study focuses on a single zoo exhibit, so further research is necessary to determine the generalizability of these results. The presence of a young giraffe calls into question whether the change in visitor perception of animal welfare is due to density alone or perhaps the appearance of family structure. COVID-19 restrictions meant that zoo occupancy (and perhaps guest comfort levels) varied across the survey period.

## 5. Conclusions

The findings of this research indicate that guest perceptions of animal welfare are connected to the degree to which positive emotions are felt while observing zoo animals, and that exhibit size and the number of animals on exhibit contribute to judgements of animal welfare. Debate around the welfare of animals in zoos should not be seen simply as a challenge to public support of zoos but perhaps more importantly as an opportunity to engage the public in meaningful dialogue about what constitutes fair treatment of animals in human care, as well as in the wild. Modelling excellent welfare, communicating best welfare practices, and understanding the reception of welfare communication amongst the zoo audience may improve mission driven learning and conservation outcomes and lead to improved welfare for animals inside and outside of zoo gates.

**Author Contributions:** Conceptualization, W.C.S., D.E.M., C.N.M. and M.L.M.; methodology W.C.S., D.E.M., C.N.M. and M.L.M.; software W.C.S., C.N.M. and M.L.M.; formal analysis, W.C.S. and C.N.M.; data curation, W.C.S. writing—original draft preparation, W.C.S.; writing—review and editing, W.C.S., C.N.M. and D.E.M.; visualization, W.C.S.; supervision, D.E.M., C.N.M. and M.L.M. All authors have read and agreed to the published version of the manuscript.

**Funding:** This research received no external funding.

**Institutional Review Board Statement:** The study was conducted in accordance with the Declaration of Helsinki, and all procedures were approved by the University of California, Berkeley Office for Protection of Human Subjects (CPHS 2021-03-14178). COVID-19 safety protocols were approved by the Vice Chancellor for Research Office at the University of California, Berkeley.

**Informed Consent Statement:** Informed consent was obtained from all subjects involved in the study.

**Data Availability Statement:** Data available upon request.

**Acknowledgments:** Thank you to research assistants Tesla Sampson, Tory Parkinson, Holly Masterson, Colby Smith, and Chloe Lujan for their support in data collection. Thanks also to Jerry Luebke and Lance Miller for early guidance and permission to use the Brookfield Internal Visitor Predisposition Survey. Gratitude to Lawrence Cohen for invaluable support with the IRB process. Thank you to the anonymous reviewers whose recommendations strengthened our results. Thanks also to Ann Marie Bisagno, Stacie Picone, and Oakland Zoo education and animal care staff.

**Conflicts of Interest:** The authors declare no conflict of interest.

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
