# Peer review of "The More the Merrier? Zoo Visitors’ Affective Responses and Perceptions of Welfare across an Increase in Giraffe Density"

_2673-5636, doi:10.3390/jzbg3020023_

Round 1
Reviewer 1 Report
In the title of the article, the authors ask the question more better, it does not find an unambiguous answer in the manuscript, it should be considered whether it is necessary. The application suggests the need to simplify the assessments and does not fully complete the results, I suggest redrafting it and linking the conclusion directly with the results and purpose of the work included in the research questions.
The authors' manuscript devoted a lot of time to the development and presentation of a subjective rating scale for visitors. It wasn't described whether the visitors had a problem with defining such strong emotions as love in such a short evaluation time (3-5 min as indicated by the authors)?
The presentation of the results seems monotonous, and the emerging statistical relationships have not been highlighted sufficiently, I recommend presenting some of the tables in the form of charts, and discussing the statistically significant relationships more broadly.
Author Response
Response to Reviewer 1 Comments
Point 1: In the title of the article, the authors ask the question more better, it does not find an unambiguous answer in the manuscript, it should be considered whether it is necessary. The application suggests the need to simplify the assessments and does not fully complete the results, I suggest redrafting it and linking the conclusion directly with the results and purpose of the work included in the research questions.
Response 1: We agree that the question of how the change in giraffe density influenced guest perceptions was insufficiently highlighted in the paper. We have altered the discussion section to emphasize these results, and added a figure to demonstrate them visually.
Point 2: The authors' manuscript devoted a lot of time to the development and presentation of a subjective rating scale for visitors. It wasn't described whether the visitors had a problem with defining such strong emotions as love in such a short evaluation time (3-5 min as indicated by the authors)?
Response 2: It would be very interesting to assess visitors' affective responses using more qualitative methods in future work, but in this case we chose an affective response instrument successfully used in similar studies of visitors.
Point 3: The presentation of the results seems monotonous, and the emerging statistical relationships have not been highlighted sufficiently, I recommend presenting some of the tables in the form of charts, and discussing the statistically significant relationships more broadly.
Response 3: We have expanded on our results in the discussion section and created a new figure to demonstrate the responses to the animal welfare perception scale visually. Thank you for your constructive comments, we feel that these have greatly improved the manuscript.
Reviewer 2 Report
Page 2; line 88: The question is not numbered.
Page 3; line 95: Mention the months in the brackets after the seasons. This is because on page 3; lines 117 to 118, months of data collection are mentioned. This will maintain consistency in description of time period for data collection.
Page 3; line 96 to 97: Are the periods after and before introduction of giraffes written here same as phase one and phase two mentioned on page 3; lines 117 to 118? If yes, then please mention so. Otherwise, it is confusing to understand the connection between ‘periods before and after’ and ‘phase one and two’. It is better to use consistent terms.
Page 4; line 179: Please explain VPS scale. Is it same as ‘Visitor’s Predisposition to Nature Scale’? If yes, then: a. there is a mismatch data items mentioned on page 4; line 178 (it says 7 including composite score) and the table 1 (table shows 8 items + composite score); and b. Should not it be VPN scale?
Page 4; Table 1: This table gives the data and multiple tests done using the data. Since multiple tests are done here on the data obtained on the same set of individuals, a correction to control Type I error (like Bonferroni) is needed.
Discussion section: The conclusions obtained from the tests results mentioned in table 1 are not clear in the discussion session, especially for the ‘Visitor’s Predisposition to Nature Scale’. Why some of the items showed significant increase with increased density of Giraffes needs to be explained clearly.
Author Response
Response to Reviewer 2
Point 1: Page 2; line 88: The question is not numbered.
Response 1: Thank you for catching this. We have numbered the question.
Point 2: Page 3; line 95: Mention the months in the brackets after the seasons. This is because on page 3; lines 117 to 118, months of data collection are mentioned. This will maintain consistency in description of time period for data collection.
Response 2: We have added the months in parenthesis after the seasons.
Point 3: Page 3; line 96 to 97: Are the periods after and before introduction of giraffes written here same as phase one and phase two mentioned on page 3; lines 117 to 118? If yes, then please mention so. Otherwise, it is confusing to understand the connection between ‘periods before and after’ and ‘phase one and two’. It is better to use consistent terms.
Response 3: We have specified that the period before the introduction of giraffes will be referred to as ‘phase one’ and that the period after the giraffes were introduced will be referred to as ‘phase two’.
Point 4: Page 4; line 179: Please explain VPS scale. Is it same as ‘Visitor’s Predisposition to Nature Scale’? If yes, then: a. there is a mismatch data items mentioned on page 4; line 178 (it says 7 including composite score) and the table 1 (table shows 8 items + composite score); and b. Should not it be VPN scale?
Response 4: Yes, VPS is the same scale initially referred to as the ‘Visitor Predisposition to Nature Scale’ in the manuscript.
a. We have corrected the mistake, there are 8 items.
b. The original name of the scale is the ‘Visitor Predisposition Scale’, and we have adjusted the manuscript so that the scale is introduced as the Visitor Predisposition Scale (VPS) and elaborated on the purpose of the scale to avoid confusion.
Point 5: Page 4; Table 1: This table gives the data and multiple tests done using the data. Since multiple tests are done here on the data obtained on the same set of individuals, a correction to control Type I error (like Bonferroni) is needed.
Response 5: Thank you for this recommendation. We adjusted p-values using the Bonferroni adjustment, which altered our results. We have incorporated these new results into our discussion.
Point 6: Discussion section: The conclusions obtained from the tests results mentioned in table 1 are not clear in the discussion session, especially for the ‘Visitor’s Predisposition to Nature Scale’. Why some of the items showed significant increase with increased density of Giraffes needs to be explained clearly.
Response 6: We have expanded on the results shown in table 1 in our discussion. After the Bonferroni adjustment, the only item which showed a significant increase was the first item, and we have addressed this in the discussion. We feel that this feedback has significantly improved the quality of the manuscript.
Reviewer 3 Report
Present investigation is very good. This type of research will helpful in wild animals safety along with the education about these animals while the people visits Zoo.
Author Response
Response to Reviewer 3:
Point 1: Present investigation is very good. This type of research will helpful in wild animals safety along with the education about these animals while the people visits Zoo.
Response 1: Thank you very much for this positive feedback.